# Gradient-EM Bayesian Meta-learning

**Yayi Zou**
Didi AI Labs @Silicon Valley
`yz725@cornell.edu`

**Xiaoqi Lu**
Columbia University
`lx2170@columbia.edu`

## Abstract

Bayesian meta-learning enables robust and fast adaptation to new tasks with uncertainty assessment. The key idea behind Bayesian meta-learning is empirical Bayes inference of hierarchical model. In this work, we extend this framework to include a variety of existing methods, before proposing our variant based on gradient-EM algorithm. Our method improves computational efficiency by avoiding back-propagation computation in the meta-update step, which is exhausting for deep neural networks. Furthermore, it provides flexibility to the inner-update optimization procedure by decoupling it from meta-update. Experiments on sinusoidal regression, few-shot image classification, and policy-based reinforcement learning show that our method not only achieves better accuracy with less computation cost, but is also more robust to uncertainty.

## 1 Introduction

Meta-learning, also known as *learning to learn*, has gained tremendous attention in both academia and industry, especially with applications to few-shot learning[3]. These methods utilize the similar nature of multi-task setting, such that learning from previous tasks helps mastering new tasks faster.

The early fast meta-learning algorithm was gradient-based and deterministic, which may cause overfitting on both inner-level and meta-level [15]. With growing interests in prediction uncertainty evaluation and overfitting control, later studies explored probabilistic meta-learning methods [6, 27, 4]. It has been agreed that Bayesian inference is one of the most convenient choices because of its Occam's Razor property [14] that automatically prevents overfitting, which happens in deep neural network (DNN) very often. It also provides reliable predictive uncertainty because of its probabilistic nature. This makes Bayesian methods important to DNN, which as [8] shows, unlike shallow neural networks, are usually poorly calibrated on predictive uncertainty.

The theoretical foundation of Bayesian meta-learning is hierarchical Bayes (HB) [5] or empirical Bayes (EB) [22], where the former can be seen as adding a hyper-prior over the latter [20]. For simplicity, in this paper we focus on EB to restrict the learning of meta-parameters to point estimates. A common solution of EB is a bi-level iterative optimization procedure [20, 13], where the "inner-update" refers to adaptation to a given task, and the "meta-update" is the meta-training objective. Starting with a generalized meta learning setting, we propose our general Bayesian framework and claim its optimality under certain metrics. This framework extends the original optimization framework for train/val split in the inner-update procedure to mitigate in-task overfitting which is important for NN based ML. We also hypothesize a mechanism of how EB framework achieves fast-adaptation (few inner-update gradient steps) under Gaussian parameterization, along with empirical evidences. What's more, we successfully adapt this EB framework to RL both theoretically and empirically which has not been done before.

We show that many important previous works in (Bayesian) meta-learning [20, 4, 27, 3, 17] can be included to this extended framework. However in these previous works, the meta-update step requires backpropagation through the inner optimization process [18] which imposes large computation

and memory burden as the increase of inner-update gradient steps. This puts limits on possible applications, especially those require many inner-update gradient steps or involves large dataset (Appendix C.2). Motivated by the above observations, we propose a gradient-based Bayesian algorithm inspired by Gradient-EM algorithm. By designing a new way to compute gradient of meta loss in Bayesian MAML, we come up with an algorithm that decouples meta-update and inner-update and thus avoids the computation and memory burden of previous methods, making it scalable to a large number of inner-update gradient steps. In addition, it enables large flexibility on the choice of inner-update optimization method because it only requires the value of the result of the inner-update optimization, instead of the optimization process (for example in experiments we use Adam in classifications and Trust Region Policy Optimization in RL). The separability of meta-update and inner-update also makes it a potentially useful scheme for distributed learning and private learning.

In experiments, we show our method can quickly learn the knowledge and uncertainty of a novel task with less computation burden in sinusoidal regression, image classification, and reinforcement learning.

## 2 Problem Formulation and Framework

### 2.1 General Meta-learning Setting

We set up the K-shot meta-learning framework upon reinforcement learning(RL) with episode length $H$ as in [3], where supervised learning is a special case with $H = 1$. With a decision rule (policy) $f$ we can sample rollout data $D = \{x_t, a_t, r_t\}_{t=1}^{H}$ from the task environment. A decision rule (policy) $f$ can be evaluated on $D$ with loss function $\mathcal{L}(D, f)$. We assume each task $\tau$ to be i.i.d. sampled from the task space $\mathcal{T}$, following some task distribution $P(\tau)$. During meta-training phase we are given a set of training tasks $\mathcal{T}^{\text{meta-train}}$. For each task $\tau'$ of this set, we collect $K$ samples rollout of current policy $f$ denoted as $D_{\tau'}^{\text{tr}}$, and another $K$ samples rollout after 1 policy gradient training of $f$ denoted as $D_{\tau'}^{\text{val}}$ ($f$ is not needed in generating samples in supervised learning). At meta-testing phase, for a randomly sampled task $\tau$, $D_{\tau}^{\text{tr}}$ is firstly provided. We are then required to return $f_\tau$ based on $\{D_{\tau'}^{\text{tr}} \bigcup D_{\tau'}^{\text{val}} : \tau' \in \mathcal{T}^{\text{meta-train}}\} \cup D_{\tau}^{\text{tr}}$ and evaluate its expected loss $l_\tau = E_{D_\tau \sim \tau}\mathcal{L}(D_\tau, f_\tau)$ on more samples generated from that task. The objective is to come up with an algorithm that produces a decision rule $f_\tau$ that minimize the expected test loss over all tasks $E_{\tau \sim P(\tau)} l_\tau$.

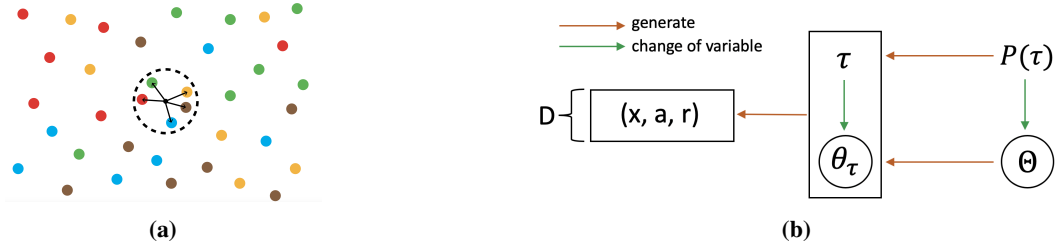

**(a)**                                                            **(b)**

**Figure 1:** (a) Minimal $A^\star$. Each dot (in a vector space) represents the weights making the NN model perform well for certain task. Different colors are used to represent different tasks. This figure demonstrates that many good solutions(local optimums of NN loss function) exist for each task. The area within dotted circle is the small neighboring zone $A^*$ where each color has at least one point inside. (b) Graphical Model

### 2.2 Extended Empirical Bayesian Meta-learning Framework

We consider parameterized decision rule $f_\theta$ and construct a corresponding generative model $\log P(D|\theta) = -\mathcal{L}(D, f_\theta)$ (We leave the detail of this construction in RL to Appendix B.1). For each task $\tau$, denote the best policy parameter as well as the best fitted underlying generative model parameter to be $\theta_\tau = \arg\min E_{D_\tau \sim \tau} \mathcal{L}(D_\tau, f_\theta) = \arg\max E_{D_\tau \sim \tau} \log P(D_\tau|\theta)$. In general such maximum is not unique, which is discussed in Section 2.3. With $\theta_\tau$ uniquely defined, we have a distribution $P(\theta_\tau)$ induced by $P(\tau)$ (change of variable). We summarize the graphical model in Figure 1(b). Under perfect approximation, the ground-truth generator matches our generative model: $P(D_\tau|\tau) = P(D_\tau|\theta_\tau)$, resulting in the following proposition (proof in Appendix A.1):

**Proposition 1.** *Suppose a data generator is represented by the hierarchical model $P(D_\tau|\theta_\tau)$ and $P(\theta_\tau)$, and define $L(Q; D) = \log E_{\theta \sim Q} P(D|\theta)$ for distribution $Q$ over $\theta$. Let $(D_\tau^{tr}, D_\tau^{val})$ be independent*

*samples from task $\tau$, and consider $Q$ determined by $D_\tau^{tr}$ via $Q = g(D_\tau^{tr})$. Then*

$$P(\theta_\tau|D_\tau^{tr}; P(\theta_\tau)) = \arg\max_g E_\tau \, L(g(D_\tau^{tr}), D_\tau^{val}) \tag{1}$$

*where $P(\theta_\tau|D_\tau^{tr}; P(\theta_\tau))$ is the posterior given the prior as $P(\theta_\tau)$ and observations $D_\tau^{tr}$.*

Two observations are made here. First, this theorem guarantees the best decision rule we can come up with during meta-testing, i.e., through computing posterior $P(\theta_\tau|D_\tau^{tr}; P(\theta_\tau))$. Second, this theorem suggests an estimation method for $P(\theta_\tau)$ during meta-training: $\arg\max_{P(\theta_\tau)} \sum_\tau L(P(\theta_\tau|D_\tau^{tr}; P(\theta_\tau)); D_\tau^{val})$. We prove in Appendix A.2 that this estimator is not only asymptotic consistent but also with good asymptotic normality which means it quickly converge to true value with small variance as number of tasks increases. We further parameterize $P(\theta_\tau)$ by $P(\theta_\tau; \Theta)$, and introduce short notation $L(\Theta, D_\tau^{tr}; D_\tau^{val}) = L(P(\theta_\tau|D_\tau^{tr}, \Theta); D_\tau^{val})$, then the optimization in meta-training can be written as $\arg\max_\Theta \sum_\tau L(\Theta, D_\tau^{tr}; D_\tau^{val})$. For clarity we denote $L_\tau^{[1]} = L(\Theta; D_\tau^{tr} \cup D_\tau^{val})$ and $L_\tau^{[2]} = L(\Theta, D_\tau^{tr}; D_\tau^{val})$, and also $L^{[1]} = E_\tau L_\tau^{[1]}$, $L^{[2]} = E_\tau L_\tau^{[2]}$. Then the estimation method of $P(\theta_\tau; \Theta)$ becomes $\arg\max_\Theta L^{[2]}$. This is an extension of the popular MLE approach in empirical Bayes, which maximize (marginal) log-likelihood $L^{[1]}$ as the special case of $L^{[2]}$ when $|D_\tau^{tr}| = 0$. There is a bias/variance trade-off between $L^{[1]}$ and $L^{[2]}$. Using $L^{[2]}$ as the meta loss function improves in-task overfitting problem while $L^{[1]}$ extracts more information from the data.

Combining the above two observations, a stochastic gradient descent (SGD) approach to meta-training is provided in Algorithm 1: at iteration $t$, gradient $\nabla_\Theta L_\tau^{[i]}$ for each task in the $t$-th meta-training batch is computed by subroutine `Meta-Gradient`, then gradient ascent on $\Theta$ is performed. A variational inference (VI) approach to meta-testing is also included, where posterior $P(\theta_\tau|D_\tau^{tr}; \Theta)$ is estimated with fixed $\hat\Theta$ learned during meta-training. Detailed discussion of these subroutines are presented in Session 3.

## 2.3   non-uniqueness and fast-adaptation

For neural networks $f_\theta$, there exists many local optimums that achieves similarly good performance for each task. We observe from empirical study (Appendix C.3) that the key to fast-adaptation for gradient-based algorithm is to find a small neighbouring zone $A^\star$ where most tasks have at least one good parameter inside it (Figure 1(a)). The intuition is that when $\{\theta_\tau\}$ are close enough they can be learned within a few gradient steps starting from any points within that neighbouring zone (our experiment shows that a perturbation of initial points within that area would still have good performance at meta-test). The existence of this small neighbouring zone $A^\star$ depends on the parametric model $f_\theta$ and the task distribution $P(\tau)$. We further demonstrate (Appendix C.3) its existence with neural networks as the parametric model and Gaussian parameterization of $P(\theta_\tau; \Theta)$ for uni-modal task distribution like sinusoidal functions. Even if we fail to find a single small neighbouring zone $A^\star$ (e.g. multi-modal task distribution like mixture of sinusoidal, linear and quadratic functions), solution may be provided by extension to mixture Gaussian [7, 19]. In this work we focus on the uni-modal situation and leave the extension to future work.

```
1  Algorithm Meta-train()
2  |   randomly initialize Θ
3  |   t = 0
4  |   while not done do
5  |   |   Sample batch of tasks 𝒯_t ∼ P(τ)
6  |   |   for each task τ ∼ 𝒯_t do
7  |   |   |   Sample D_τ^tr, D_τ^val ∼ τ
8  |   |   |   Compute ∇_Θ L_τ^[i] by Subroutine
   |   |   |     Meta-Gradient(Θ^(t),{D_τ^tr, D_τ^val})
9  |   |   end
10 |   |   Θ^(t+1) = Θ^(t) − β ∑_{τ∈𝒯_t} ∇_Θ L_τ^[i]
11 |   |   t = t + 1
12 |   end
```

```
1  Algorithm Meta-test()
2  |   Require: learned Θ, D_τ^tr from new task τ
3  |   Compute posterior λ_τ =VI(Θ, D_τ^tr).
4  |   Sample θ_τ ∼ P(θ; λ_τ)
5  |   return f(; θ_τ) for evaluation
1  Subroutine VI(Θ,D_τ)
2  |   Initialize λ_τ at Θ.
3  |   while not done do
4  |   |   Sample ε̃ from ε ∼ p(ε).
5  |   |   λ_τ = λ_τ + α∇_{λ_τ}[log P(D_τ|g(λ_τ, ε̃))
6  |   |   - KL(P(θ_τ; λ_τ) ‖ P(θ_τ|Θ))]
7  |   end
8  |   return λ_τ
```

**Algorithm 1:** Extended Empirical Bayes Meta-learning Framework. VI: reparameterize $\tilde\theta_\tau \sim P(\theta_\tau; \lambda_\tau)$ using a differentiable transformation $g(\lambda_\tau, \epsilon)$ of an auxiliary noise variable $\epsilon$ such that $\tilde\theta_\tau = g(\lambda_\tau, \epsilon)$ with $\epsilon \sim p(\epsilon)$ [12]

## 3 Method

In this section, we first introduce the gradient-based variational inference subroutine `VI` related to a variety of existing methods, then present our proposed subroutine `Meta-Gradient` inspired by Gradient-EM algorithm and compare it with the mostly used existing methods for this subroutine.

### 3.1 Variational Inference

Notice that this framework requires computing posterior on complex models such as neural networks. To achieve this, we approximate the posterior with the same parametric distribution $P(\theta_\tau; \lambda_\tau)$ as we approximate the prior $P(\theta_\tau; \Theta)$ and use Variational Inference to compute the parameters, as has been done in previous work [20]. Let $P(\theta_\tau; \lambda_\tau(D_\tau; \Theta))$ be the approximation of the posterior $P(D_\tau, \theta_\tau; \Theta))$ by minimizing their KL distance. Since

$$L(\Theta; D_\tau) = \log P(D_\tau; \Theta) = KL[P(\theta_\tau; \lambda_\tau) \parallel P(\theta_\tau|D_\tau; \Theta)] + E_{P(\theta_\tau; \lambda_\tau)}[\log P(D_\tau, \theta_\tau; \Theta) - \log P(\theta_\tau; \lambda_\tau)] \quad (2)$$

is constant in terms of $\lambda_\tau$, we have $\lambda_\tau(D_\tau; \Theta) = \arg\min_{\lambda_\tau} KL[P(\theta_\tau; \lambda_\tau) \parallel P(\theta_\tau|D_\tau; \Theta)] = \arg\max_{\lambda_\tau} E_{P(\theta_\tau; \lambda_\tau)}[\log P(D_\tau, \theta_\tau; \Theta) - \log P(\theta_\tau; \lambda_\tau)] = \arg\max_{\lambda_\tau} E_{P(\theta_\tau; \lambda_\tau)}[\log p(D_\tau|\theta_\tau)] - KL[P(\theta_\tau; \lambda_\tau) \parallel p(\theta_\tau; \Theta)]$. So the inference process is to find $\lambda_\tau$ to maximize the Evidence Lower Bound $ELBO^{(\tau)}(\lambda_\tau; \Theta) = E_{P(\theta_\tau; \lambda_\tau)}[\log p(D_\tau|\theta_\tau)] - KL[P(\theta_\tau; \lambda_\tau) \parallel p(\theta_\tau; \Theta)]$ via mini-batch gradient descent[20]. The gradient of KL-divergence terms are calculated analytically in Gaussian case whereas the gradient of expectations can be computed by monte-carlo with reparameterization along with some variance reduction tricks [11, 28]. Due to the above analysis in Section 2.3, only a few gradient steps are needed for this process with well learned $\Theta$ by our framework. We summarize the subroutine `VI` in Algorithm 1.

A special case worth mentioning is when we use delta function for the posterior approximation $P(\theta_\tau; \lambda_\tau) = \delta_{\mu_{\lambda_\tau}}(\theta_\tau)$, we have $\lambda_\tau(D_\tau; \Theta) = \arg\max[\log P(D_\tau|\mu_{\lambda_\tau}) - \parallel \mu_{\lambda_\tau} - \mu_\Theta \parallel^2 / (2\Sigma_\Theta^2)]$, which is actually the inner-update step of iMAML [18], MAML [3], and reptile [17] (if we replace the l2 regularization term with choosing $\mu_\Theta$ as initial point for gradient based optimization: $\mu_{\lambda_\tau}(\mu_\Theta) = \mu_\Theta - \nabla_\theta \log P(D_\tau|\theta)|_{\theta=\mu_\Theta}$).

### 3.2 Meta-Gradient

The essential part of this meta-learning framework is to compute the gradient $\nabla_\Theta L_\tau^{[i]}$. We show below that this problem can be reduced as computing $\nabla_\Theta L(\Theta; D_\tau) = \nabla_\Theta \log P(D_\tau; \Theta)$ given $D_\tau$ and $\Theta$. For $L^{[1]}$, this is direct. For $L^{[2]}$, there are two approaches. The first approach is to compute $\lambda_\tau(D_\tau; \Theta)$ as stated above, then

$$\nabla_\Theta L_\tau^{[2]} = \nabla_\Theta L(\Theta, D_\tau^{\text{tr}}; D_\tau^{\text{val}}) = \nabla_\Theta L(\lambda_\tau(D_\tau^{\text{tr}}; \Theta); D_\tau^{\text{val}}) = \nabla_\Theta L(\Theta; D_\tau^{\text{val}})|_{\Theta=\lambda_\tau(D_\tau^{\text{tr}}; \Theta)} * \nabla_\Theta \lambda_\tau(D_\tau^{\text{tr}}; \Theta) \quad (3)$$

, where $\nabla_\Theta \lambda_\tau(D_\tau^{\text{tr}}; \Theta)$ can be computed by auto-gradient (if $\lambda_\tau(D_\tau^{\text{tr}}; \Theta)$ is computed by gradient based algorithms). This approach is widely used in previous work such as [3], [4], [27], [6]. The second approach is proposed by us as shown in subroutine `Meta-Gradient:GEM-BML+` below. We utilize a property $L(\Theta, D_\tau^{\text{tr}}; D_\tau^{\text{val}}) = L(\Theta; D_\tau^{\text{tr}} \bigcup D_\tau^{\text{val}}) - L(\Theta; D_\tau^{\text{tr}})$ (proof in Appendix A.4) such that $\nabla_\Theta L_\tau^{[2]} = \nabla_\Theta L(\Theta; D_\tau^{\text{tr}} \bigcup D_\tau^{\text{val}}) - \nabla_\Theta L(\Theta; D_\tau^{\text{tr}})$ can be expressed by the difference of two $L^{[1]}$ terms, thus is reduced to computing $\nabla_\Theta L(\Theta; D_\tau)$ terms.

**Gradient-EM Estimator**
We propose an efficient way to compute $\nabla_\Theta L(\Theta; D_\tau)$ through gradient of the *complete* log likelihood. This is guaranteed by the following Gradient-EM Theorem inspired by the observation in [23].

**Theorem 1.** $\nabla_\Theta L(\Theta; D) = E_{\theta \sim P(\theta|D; \Theta)} \nabla_\Theta \log[P(D, \theta; \Theta)]$

*Proof.*

$$\nabla_{\Theta} L(\Theta; D) = \frac{\partial}{\partial \Theta} \log P(D|\Theta)$$

$$= \frac{1}{P(D|\Theta)} \frac{\partial}{\partial \Theta} \int P(D, \theta|\Theta) d\theta$$

$$= \int \frac{P(D, \theta|\Theta)}{P(D|\Theta)} \frac{\partial}{\partial \Theta} \log P(D, \theta|\Theta) d\theta$$

$$= \int P(\theta|D, \Theta) \frac{\partial}{\partial \Theta} \log P(D, \theta|\Theta) d\theta$$

$$= E_{\theta \sim P(\theta|D;\Theta)} \nabla_{\Theta} \log[P(D, \theta; \Theta)]$$

$\square$

Under hierarchical modeling structure, we have $\nabla_{\Theta} \log P(D_{\tau}, \theta_{\tau}|\Theta) = \nabla_{\Theta} \log[P(D_{\tau}|\theta_{\tau}) * P(\theta_{\tau}; \Theta)] = \nabla_{\Theta} \log[P(\theta_{\tau}; \Theta)]$. Combining with Theorem 1 we have $\nabla_{\Theta} L(\Theta; D_{\tau}) = E_{\theta_{\tau} \sim P(\theta|D_{\tau};\Theta)} \nabla_{\Theta} \log[P(\theta_{\tau}; \Theta)]$. After using VI to compute the approximate posterior parameter $\lambda_{\tau}(D_{\tau}, \Theta)$, the above estimator becomes $\hat{g} = E_{\theta_{\tau} \sim P(\theta_{\tau}; \lambda_{\tau}(D_{\tau}, \Theta))} \nabla_{\Theta} \log[P(\theta_{\tau}; \Theta)]$ which can be calculated analytically in Gaussian case as we show in Appendix B.7. This gives us two `Meta-Gradient` subroutines GEM-BML and GEM-BML+ for $\nabla_{\Theta} L_{\tau}^{[1]}$ and $\nabla_{\Theta} L_{\tau}^{[2]}$ respectively. We name Algorithm 1 with these two subroutines as our algorithms GEM-BML and GEM-BML+.

1 **Subroutine**
  `Meta-Gradient:GEM-BML(`$\Theta, \{D^{tr}, D^{val}\}$`)`
2      Compute posterior $\lambda^{tr} =$`VI`$(\Theta, D^{tr})$.
3      Compute posterior $\lambda^{tr \oplus val} =$`VI`$(\lambda^{tr}, D^{val})$.
4      return $\hat{g} = E_{\theta \sim P(\theta; \lambda^{tr \oplus val})} \nabla_{\Theta} \log[P(\theta; \Theta)]$

1 **Subroutine**
  `Meta-Gradient:GEM-BML+(`$\Theta, \{D^{tr}, D^{val}\}$`)`
2      Compute posterior $\lambda^{tr} =$`VI`$(\Theta, D^{tr})$.
3      Compute posterior $\lambda^{tr \oplus val} =$`VI`$(\lambda^{tr}, D^{val})$.
4      return
      $\hat{g} = E_{\theta \sim P(\theta; \lambda^{tr \oplus val})} \nabla_{\Theta} \log[P(\theta; \Theta)] -$
      $E_{\theta \sim P(\theta; \lambda^{tr})} \nabla_{\Theta} \log[P(\theta; \Theta)]$

**ELBO Gradient Estimator**
As comparison, one of the most widely used methods to optimize $L^{[i]}$ in Bayesian meta-learning is optimizing ELBO [20](see Appendix B.6 for other existing methods and comparing analysis). Here we show it is actually another way to estimate $\nabla_{\Theta} L(\Theta; D_{\tau})$. According to equation (2), when VI approximation error $KL[P(\theta_{\tau}; \lambda_{\tau}(D_{\tau}; \Theta)) \parallel P(\theta_{\tau}|D_{\tau}; \Theta)]$ is small enough, we have $L(\Theta; D_{\tau}) \simeq E_{P(\theta_{\tau}; \lambda_{\tau}(D_{\tau};\Theta))}[\log P(D_{\tau}, \theta_{\tau}; \Theta) - \log P(\theta_{\tau}; \lambda_{\tau}(D_{\tau};\Theta))] = [E_{P(\theta_{\tau}; \lambda_{\tau}(D_{\tau};\Theta))} \log P(D_{\tau}|\theta_{\tau})] - KL[P(\theta_{\tau}; \lambda_{\tau}(D_{\tau}; \Theta)) \parallel P(\theta_{\tau}|\Theta)] = ELBO^{(\tau)}(\lambda_{\tau}(D_{\tau}; \Theta); \Theta)$. So the gradient can be computed by

$$\nabla_{\Theta} L(\Theta; D_{\tau}) \simeq \nabla_{\Theta} ELBO^{(\tau)}(\lambda_{\tau}(D_{\tau}; \Theta); \Theta)$$
$$= \frac{\partial}{\partial \lambda_{\tau}} ELBO^{(\tau)}(\lambda_{\tau}; \Theta)|_{\lambda_{\tau} = \lambda_{\tau}(D_{\tau};\Theta)} * \nabla_{\Theta} \lambda_{\tau}(D_{\tau}; \Theta) + \frac{\partial}{\partial \Theta} ELBO^{(\tau)}(\lambda_{\tau}; \Theta)|_{\lambda_{\tau} = \lambda_{\tau}(D_{\tau};\Theta)} \quad (4)$$

. The first partial gradient term can be computed by the same method in Section 3.1 and the second one can be calculated analytically in Gaussian case.

In fact, Gradient-EM(GEM) can also be reviewed as an co-ordinate descent algorithm to optimize ELBO as a variant of EM as we show in Appendix B.3. Comparing to ELBO gradient, GEM avoids the backProps computation of $\nabla_{\Theta} \lambda_{\tau}(D_{\tau}; \Theta)$ which gives it a series of advantages as we specify in Section 4.2. Both GEM and ELBO gradient has estimation error arise from the discrepancy of estimated posterior by VI and the true posterior. We show empirical results in Appendix C.1 that GEM has stably lower estimation error than ELBO gradient. We also show in Appendix A.3 that our method has a theoretical bound of estimation error in terms of the VI discrepancy $\parallel \hat{g} - \nabla_{\Theta} L(\Theta; D_{\tau}) \parallel \leq M \sqrt{D_{KL}(P(\theta|D_{\tau}; \Theta) \parallel P(\theta; \lambda_{\tau}(D_{\tau}, \Theta)))}$ where $M$ is a bounded constant.

## 4 Analysis

**Matrix of Related Works**
We compare Gradient-EM (our method) with ELBO-gradient over two loss functions $L^{[1]}, L^{[2]}$,

| | $L^{[1]} = E_{\tau \in \mathcal{T}} L(\Theta; D_\tau^{\text{tr}} \cup D_\tau^{\text{val}})$ | $L^{[2]} = E_{\tau \in \mathcal{T}} L(\lambda_\tau(D_\tau^{\text{tr}}; \Theta); D_\tau^{\text{val}})$ |
|---|---|---|
| ELBO gradient | Amortized BML [20] | related to PMAML [4] |
| Graident-EM | GEM-BML (our method) ; reduce to Reptile [17] in delta case | KL-Chaser Loss(related to l2-Chaser Loss, BMAML [27]) |

**Table 1:** Matrix of related works

summarized in Table 1. It turns out each element of this matrix is related to a previous work or our method. Notice that this matrix can be extended with more columns (e.g. one more column of $L^{[2]} = E_{\tau \in \mathcal{T}} L(\Theta; D_\tau^{\text{tr}} \cup D_\tau^{\text{val}}) - L(\Theta; D_\tau^{\text{tr}}))$ and more rows to a larger matrix with blank elements (models) that haven't been explored before. For example, KL-Chaser Loss model in the right bottom of Table 1 hasn't been studied before. We leave the thorough study of all combinations to future work. Here we only show how MAML and Reptile can be fit into this Bayes frame, while further details are left to Appendix B.4. To see this, consider using fixed variance parameters for both prior $P(\theta_\tau; \mu_\Theta, \Sigma_\Theta = C_0)$ and posterior $P(\theta_\tau; \mu_{\lambda_\tau}, \Sigma_{\lambda_\tau} = C_\tau)$ and let $C_\tau \to 0$ so posterior becomes delta distribution $\delta_{\mu_{\lambda_\tau}}(\theta_\tau)$. We can compute $\mu_{\lambda_\tau}(\mu_\Theta)$ by gradient descent from $\mu_\Theta$. MAML uses $L^{[2]}$ as meta-loss function. Under delta distribution posterior we have

$$L_\tau^{[2]} = \log \int P(D_\tau|\theta_\tau)\delta_{\mu_{\lambda_\tau}(\mu_\Theta)}(\theta_\tau)d\theta_\tau = \log P(D_\tau|\mu_{\lambda_\tau}(\mu_\Theta)) = f(D_\tau; \mu_{\lambda_\tau}(\mu_\Theta))$$

. Then $\nabla_\Theta L^{[2]} = \nabla_{\mu_\Theta} f(D_\tau; \mu_{\lambda_\tau}(\mu_\Theta))$ can be directly computed through back-propagation in neural networks. On the other hand, Reptile uses $L^{[1]}$ as meta-loss function. Using GEM-gradient $\hat{g}$ we have

$$\nabla_\Theta L_\tau^{[1]} = E_{\delta_{\mu_{\lambda_\tau}(\mu_\Theta)}(\theta_\tau)} \nabla_\Theta \log[P(\theta_\tau; \mu_\Theta, \Sigma_\Theta = C_0)] = \nabla_{\mu_\Theta} \frac{|\mu_\Theta - \mu_{\lambda_\tau}(\mu_\Theta)|^2}{2 * C_0{}^2}$$

which is the Reptile gradient. Also notice that, if we let $C_0 \to 0$ and so the prior becomes delta, then we have

$$\nabla_\Theta L_\tau^{[1]} = \nabla_{\mu_\Theta} \log \int P(D_\tau|\theta_\tau)\delta_{\mu_\Theta}(\theta_\tau)d\theta_\tau = \nabla_{\mu_\Theta} \log P(D_\tau|\mu_\Theta) = \nabla_{\mu_\Theta} f(D_\tau; \mu_\Theta)$$

. This corresponds to "pre-train" which simply train a model to fit data of all tasks combined.

**Advantages of GEM**
Observe that all methods in the above matrix requires to compute the posterior parameters $\lambda_\tau(D_\tau; \Theta)$ first and use it to compute the sampled meta-loss function gradient $\nabla_\Theta L_\tau^{[i]}$. Following the convention of [3], we define the step of computing $\lambda_\tau(D_\tau; \Theta)$ as *inner-update* and the step of computing $\nabla_\Theta L_\tau^{[i]}$ as *meta-update*. Notice that both the $L^{[2]} = E_{\tau \in \mathcal{T}} L(\lambda_\tau(D_\tau^{\text{tr}}; \Theta); D_\tau^{\text{val}})$ column and the ELBO-gradient row involve the computation of $\nabla_\Theta \lambda_\tau(D_\tau; \Theta)$(Equation (3,4)). This means the meta-update computation of these three methods(highlighted in colour) has to compute backpropagation through the inner optimization process which leads to a number of burden and limitation, while GEM avoids this computation and thus gives a number of advantages as mentioned in Introduction. Also notice that, if assuming independence between neural network layers, the meta-update of our algorithm (Line 4 of Subroutine GEM-BML(+)) can be computed among different neural network layers in parallel, which may largely reduce the computation time in deep neural networks. We summarize a detailed analysis of our advantages to Appendix B.5.

## 5 Experiment

### 5.1 Regression

The purpose of this experiment is to test our methods on fast adaptation ability and robustness to meta-level uncertainty.

We compare our model GEM-BML and GEM-BML+ with MAML [3], Reptile [17] and Bayesian meta-learning benchmarks BMAML [27] and Amortized BML[20] on the same sinusoidal function regression problem. We first apply the default setting in [3] then apply a more challenging setting which contains more uncertainty as proposed in [27] to demonstrate the robustness to meta-level uncertainty. Data of each task is generated from $y = A\sin(wx + b) + \epsilon$ with amplitude A, frequency w, and phase b as task parameter and observation noise $\epsilon$. Task parameters are sampled from uniform distributions $A \in [0.1, 5.0], b \in [0.0, 2\pi], w \in [0.5, 2.0]$ and observation noise follows $\epsilon \sim N(0, (0.01A)^2)$. $x$ ranges from $[-5.0, 5.0]$. For each task, $K = 10$ observations($\{x_i, y_i\}$ pairs) are given. The underlying

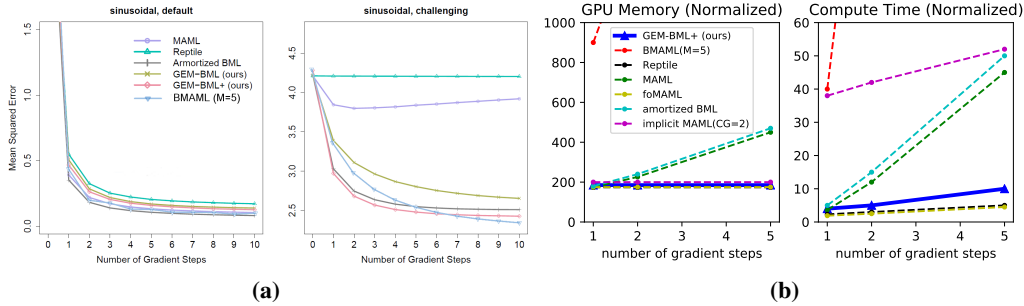

(a)                                          (b)

**Figure 2:** (a) Sinusoidal regression results: Meta-test error of default and challenging setting after 40000 meta-train iterations. (b) Computation and memory trade-offs with 4 layer CNN on 1-shot,5-class miniImageNet task. (BMAML is beyond the range of the plot.)

| Omniglot | 1-shot, 5-class | 5-shot, 5-class | 1-shot, 20-class | 5-shot, 20-class |
|---|---|---|---|---|
| MAML | $98.7 \pm 0.4$ % | $99.9 \pm 0.1$ % | $95.8 \pm 0.3$ % | $98.9 \pm 0.2$ % |
| first-order MAML | $98.3 \pm 0.5$ % | $99.2 \pm 0.2$ % | $89.4 \pm 0.5$ % | $97.9 \pm 0.1$ % |
| Reptile | $97.68 \pm 0.04$ % | $99.48 \pm 0.06$ % | $89.43 \pm 0.14$ % | $97.12 \pm 0.32$ % |
| iMAML | $99.50 \pm 0.26$ % | $99.74 \pm 0.11$ % | $96.18 \pm 0.36$ % | $99.14 \pm 0.1$ % |
| GEM-BML+(Ours) | $99.23 \pm 0.42$ % | $99.64 \pm 0.08$ % | $96.24 \pm 0.35$ % | $98.94 \pm 0.25$ % |

**Table 2:** Few-shot classification on Omniglot dataset. The $\pm$ shows 95% confidence intervals over different testing tasks. All results to compare are from original literature.

network architecture(2 hidden layers of size 40 with RELU activation) is the same as [3] to make a fair comparison.

In Figure 2 (a), we plot the mean squared error (MSE) performance on test tasks during meta-test process under both settings. Under default setting, our methods show similar fast-adaptation ability as previous methods. The challenging setting result shows that Bayesian methods GEM-BML(+), BMAML and Amortized BML can still extract information in high uncertainty environment while non-Bayesian models MAML and Reptile fail to learn. We also observe that our model provides a stable meta-train learning curve and continues to improve as performing more gradient steps without overfitting. This demonstrates the robustness of Bayesian methods resulted from its probabilistic nature and the ability to control overfitting.

## 5.2 Classification

The purpose of this experiment aims to answer the following questions: (1) Does our model save computation time and memory requirement by avoiding meta-update backProp as we claimed? (2) Can our methods be scaled to few-shot image classification benchmarks and achieve good accuracy and predictive uncertainty?

To study (1), we turn to Mini-ImageNet [21] dataset on 1-shot,5-class. We compare GEM-BML+(GEM-BML is even less expensive) with MAML and its first order variants foMAML, Reptile, iMAML, Amortized BML and BMAML in Fig 2(b). Just like other first-order meta-learning al-

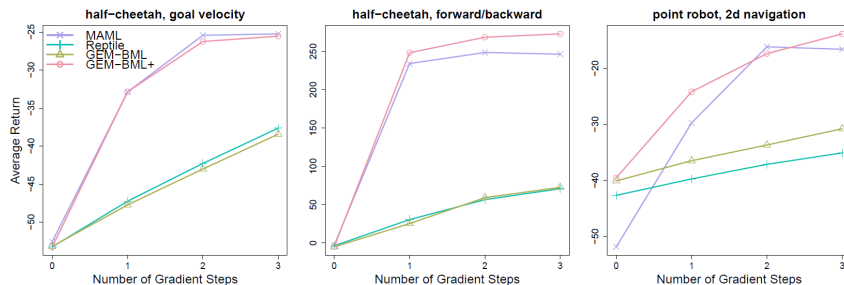

**Figure 3:** Reinforcement Learning

| miniImageNet | 1-shot, 5-class |
|---|---|
| MAML [3] | $48.70 \pm 1.84\ \%$ |
| first-order MAML [3] | $48.07 \pm 1.75\ \%$ |
| Reptile [17] | $49.97 \pm 0.32\ \%$ |
| iMAML [18] | $49.30 \pm 1.88\ \%$ |
| Amortized BML [20] | $45.0 \pm 0.60\ \%$ |
| GEM-BML+(Ours) | $50.03 \pm 1.63\ \%$ |

| Predictive uncertainty | ECE | MCE |
|---|---|---|
| MAML | 0.0471 | 0.1104 |
| Amortized BML | 0.0124 | 0.0257 |
| GEM-BML+(Ours) | 0.0102 | 0.0197 |

**Table 3:** Accuracy and Predictive Uncertainty Measurement of Few-shot classification on the MiniImagenet dataset. Small ECE and MCE indicate a model is better calibrated.

gorithms and iMAML which decouples the inner-update and meta-update, the memory usage of GEM-BML(+) is independent of the number of inner-update gradient steps since the inner-update computation other than final step results need not to be stored. On the other hand, MAML-like algorithms (MAML, Amoritized BML) need memory growing linearly with inner-update gradient steps. It is also similar for compute time, MAML-like algorithms requires expensive backProp over the inner-update optimization in meta-update, where the compute cost grows at a faster rate than GEM-BML(+), foMAML, Reptile and iMAML (iMAML has a relatively high base compute cost because of Hessian computation).

To study (2) we applied our method to N-class image classification on the Omniglot dataset and MiniImagenet dataset which are popular few-shot learning benchmarks([25, 24, 21]). Notice that the purpose of this experiment is not to compete with state-of-the-art on this benchmark but to provide an apples-to-apples comparison with prior works within our extended Empirical Bayes framework. So for a fair comparison, we use the identical backbone convolutional architecture[3] as these prior works. Note however that this backbone architecture can be replaced with other ones and lead to better results for all algorithms [2, 10]. We leave to the future work to improve our method with better backbone architectures to challenge the state-of-the-art of this benchmark. The inner-update is computed using Adam to demonstrate the flexibility of our methods in choosing inner-update optimizer. The results in Table 2 and 3 shows that our methods performs as good as the the best prior methods within our extended Empirical Bayes framework.

Predictive uncertainty is the probability associated with the predicted class label which indicates how likely the predictions to be correct. To measure the predictive uncertainty of the models, we use two quantitative metrics ECE and MCE ([16, 8]) to MiniImagenet dataset. Smaller ECE and MCE indicate a better-calibrated model. A perfectly calibrated model achieves 0 for both metrics. The results of ECE and MCE for our models and previous works are shown in Table 3. We can see that our model is slightly better calibrated compared to the state-of-art bayesian meta-learning model Amortized BML and well outperform non-Bayesian models. This shows our model can learn a good prior and make good probability predictions as an advantage of Bayesian model.

## 5.3 Reinforcement Learning

We test and compare the models on the same 2D Navigation and MuJoCo continuous control tasks as are used in [3]. See Appendix C.4.3 for detailed descriptions on experiment settings and hyper-parameters.

For a fair comparison, we use the same policy network architecture as [3] with two hidden layers, each with 100 ReLU units. At meta-train, we collect $K$ samples of rollout of current policy and another $K$ samples rollout after 1 policy gradient update as [3] where $K$ is the inner-batch size. At meta-test, we compare adaptation to a new task with up to 3 gradient updates, each with 40 samples. We compare to two baseline models: MAML and reptile.

MAML uses TRPO in meta-update to boost performance while our meta-update is data-free as specified in the above sections. For inner-updates, due to our model's flexibility of choosing inner-update optimzier, we can either use vanilla policy gradient (REINFORCE) (Williams, 1992) or a specially designed TRPO proposed by [3]. We find that TRPO inner-update performs better in 2d navigation while vanilla policy gradient tend to be better in MuJoCo continuous control tasks. We hypothesis that the reasons could be in complex task setting the task distribution variance tend to be higher($A^\star$ is larger in Figure 1 (a)). While TRPO limits the step size of each inner-update which makes the task parameters hard to be attained within a few gradient steps.

As shown in Fig 3, GEM-BML+ outperforms MAML while reptile and GEM-BML has less superior performance. This shows $L^{[2]}$ variant is necessary in RL which has high in-task variance and easily overfitted. Previous work [17] show it is hard to adapt algorithms to RL with the advantage of data-free meta-update(reptile like algorithm). But with our $L^{[2]}$ variant we can adapt to RL while preserving this advantage. Our results show that the key to adaptation is $L^{[2]}$ variant for RL, which highlighted our contribution of GEM-BML+ algorithm.

## 6  Related Works

Hierarchical Bayes(HB) and Empirical Bayes(EB) have been decently studied [9] in the past to utilize statistical connections between related tasks. Since then, deep neural network(DNN) caught enormous attention and efforts of measuring the uncertainty of DNN also started ongoing in which Bayesian and sampling method are widely applied.[1] The research trend of multi-task learning and transfer learning also changed to the fine-tuning framework for DNN after then. Model Agnostic Meta-learning(MAML) [3] emerged in such a motivation to find good initial parameters that can be fast adapted to new tasks in a few gradient steps. Recently, Bayesian models have a big comeback because of their probabilistic nature in uncertainty measure and automatic overfitting preventing. [26] applied HBM to multi-task reinforcement learning. [6] related MAML to Hierarchical Bayesian model and proposed a Laplace approximation method to capture isotopic Gaussian uncertainty of the model parameters. BMAML [27] used Stein Variational Gradient Descent(SVGD) to obtain posterior samples and proposed a Chaser Loss in order to prevent meta-level overfitting. PMAML [4] also proposed a gradient-based method to obtain a fixed measure of prior and posterior uncertainty. Amortized-BML [20] proposed a MAML-like variational inference method for amortized Bayesian meta-learning. All of the methods above can not make inner-update and meta-update separable thus largely limit the flexibility of the optimization process of inner-update. iMAML [18] propose an implicit gradients method for MAML which can make inner-update and meta-update separable with the cost of computation on second order derivatives and solving QP in each meta-update step. Notice that this method is not within our framework though it shares the style because its update is by the effort of approximating MAML rather than a Bayesian approach.

## 7  Conclusion

Inspired by Gradient-EM algorithm we have proposed GEM-BML(+) Algorithm for Bayesian Meta-learning. Our method is based on a theoretical insight of the Gradient-EM Theorem and the Bayesian formulation of multi-task meta-learning. This method avoids backProp in meta-update and decouples the meta-update and inner-update. We have tested our method on sinusoidal regression, few-shot image classifications and reinforcement learning to demonstrate the advantage of our method. For future work, we consider to apply our method to start-of-art image classification backbone and extending our work to nonparametric Gaussian approximation to handle multimodal and dynamic task-distribution situations.

## Broader Impact

Meta-learning algorithms can be applied in AI products that requires fast adaptation with few data points. Examples are: 1) facial recognition system for enterprise where only few photo shots from each employer are taken as training samples; 2) manufacturing robot that masters new tasks quickly from few times of human demonstration; 3) AI assistant that customizes to a new user after few interactions. Our research, in particular, makes an impact by introducing a novel approach that improves computational efficiency, robustness (and other advantages) of meta-learning. This generally benefits future AI researches in meta-learning, rather than direct impact on specific product.

In particular, our method improves the computational efficiency, uncertainty prediction and has potential use in building distributed and privacy protected meta-learning system. Potential advantage for deep NN because it can be parallelized among network layers. Specifically, under a distributed setting where meta-update and inner-update take place in separate devices, unlike previous methods, our method avoids transmission of gradients which may cause leakage of user data due to a recent research [29]. It may help protect user privacy and enhance decentralization of AI, preventing the monopoly of AI.

## acknowledgements

The authors would like to thank Zhiwei Qin for his support and detailed feedback on an early draft of the paper, Prof. Yuhong Guo for technical advice, Prof. Jieping Ye, Prof. Hongtu Zhu for their support, and Tristan Deleu's support on implementation and the anonymous reviewers for their comments.

The authors would also like to thank Qian Qiao for helpful support, without which this work would not be accomplished.

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
