[Supplementary Material]

# Appendix

## A. Proofs

### A.1. Theorem 1 (*Section 2.2*)

**Theorem 1.** *Suppose data generator is represented by the hierarchical model $P(D_\tau|\theta_\tau)$ and $P(\theta_\tau)$, and define $L(Q; D) = \log \mathrm{E}_{\theta \sim Q} P(D|\theta)$ for distribution $Q$ over $\theta$. Let $(D_\tau^{tr}, D_\tau^{eval})$ be independent samples from task $\tau$, and consider $Q$ determined by $D_\tau^{tr}$ via $Q = g(D_\tau^{tr})$. Then*

$$P(\theta_\tau|D_\tau^{tr}, P(\theta_\tau)) = \arg\max_g \mathrm{E}_\tau L(g(D_\tau^{tr}); D_\tau^{eval}) \tag{1}$$

*Proof.* For any distribution $Q$, observe that $\mathrm{E}_{\theta \sim Q} P(D|\theta)$ is a distribution over data $D$. For clarity of the proof, we denote $\mathrm{E}_{\theta \sim g(D_\tau^{tr})} P(D|\theta)$ by conditional distribution $P_g(D|D_\tau^{tr})$. Denote $P(D_\tau^{eval}|D_\tau^{tr}) = P(D_\tau^{eval}|D_\tau^{tr}; \Theta^\star)$ where subscript $\star$ denotes the underlying truth. Then,

$$\mathrm{E}_\tau L(g(D_\tau^{tr}); D_\tau^{eval}) = \mathrm{E}_\tau \log \mathrm{E}_{\theta \sim g(D_\tau^{tr})} P(D_\tau^{eval}|\theta) \tag{2}$$

$$= \mathrm{E}_{D_\tau^{tr}} \left[ \mathrm{E}_{D_\tau^{eval}|D_\tau^{tr}} \log P_g(D_\tau^{eval}|D_\tau^{tr}) \right] \tag{3}$$

The cross entropy term achieves maximum as $P_g(D_\tau^{eval}|D_\tau^{tr}) = P(D_\tau^{eval}|D_\tau^{tr})$, that is,

$$\int P(D_\tau^{eval}|\theta)Q(\theta)d\theta = \int P(D_\tau^{eval}|\theta)P(\theta|D_\tau^{tr})d\theta \tag{4}$$

Equation holds when $Q(\theta) = P(\theta|D_\tau^{tr})$, the posterior distribution of $\theta$ given $D_\tau^{tr}$. In other words, we have $g(D_\tau^{tr}) = P(\theta_\tau|D_\tau^{tr}, P(\theta_\tau))$, generating posterior from $D_\tau^{tr}$. Lastly, to finish the proof, note that this (point-wise) maximum is feasible, because $g$ is a function of $D_\tau^{tr}$ and the cross entropy terms is inside the expectation/integral over $D_\tau^{tr}$. □

As suggested in article, if we parameterize the prior as $P(\theta_\tau; \Theta)$, then this theorem motivates an estimator $\widehat{\Theta}$ for meta-training by empirical risk minimization ("training in the same way as testing", more explanation below):

$$\widehat{\Theta} = \arg\max_\Theta \sum_\tau L\left(P(\theta_\tau|D_\tau^{tr}, \Theta); D_\tau^{eval}\right) \tag{5}$$

### A.2. Asymptotic consistency and normality of $L^{[2]}$ estimator (*Section 2.2*)

Similar to MLE ($L^{[1]}$ case), this is also an M-estimator, thus under some regularity conditions [Van der Vaart, A. (1998). Asymptotic Statistics. Cambridge University Press.] we can establish asymptotic normality for $\widehat{\Theta}$:

$$\sqrt{n}(\widehat{\Theta}_n - \Theta^\star) \to_d \mathcal{N}\left(0, R^{-1}SR^{-1}\right) \tag{6}$$

where $n$ is the number of tasks in meta-training set, and

$$R = \mathrm{E}_\tau \left( \frac{\partial^2}{\partial\Theta\partial\Theta^T} L\left(P(\theta_\tau|D_\tau^{tr}, \Theta); D_\tau^{eval}\right)\Big|_{\Theta^\star} \right) \tag{7}$$

$$S = \mathrm{E}_\tau \left[ \left( \frac{\partial}{\partial\Theta} L\left(P(\theta_\tau|D_\tau^{tr}, \Theta); D_\tau^{eval}\right)\Big|_{\Theta^\star} \right) \left( \frac{\partial}{\partial\Theta} L\left(P(\theta_\tau|D_\tau^{tr}, \Theta); D_\tau^{eval}\right)\Big|_{\Theta^\star} \right)^T \right] \tag{8}$$

As comparison, in the MLE case, $S = -R = I(\Theta^\star)$ is Fisher information, thus the asymptotic variance is given by $I(\Theta^\star)^{-1}$, which is also the Cramer-Rao lower bound. For our $L^{[2]}$ case, first we use the Corollary: $L\left(P(\theta_\tau|D_\tau^{tr}, \Theta); D_\tau^{eval}\right) = L(\Theta; D_\tau^{tr}, D_\tau^{eval}) - L(\Theta; D_\tau^{tr})$. Then, with some calculation we obtain

$$R^{-1}SR^{-1} = \frac{m}{m-k}I(\Theta^\star)^{-1} \tag{9}$$

where $|D_\tau^{tr}| = k$ and $|D_\tau^{eval}| = m - k$. This suggests that the asymptotic variance of $L^{[2]}$ is larger than that of $L^{[1]}$ (lower efficiency) unless $k = 0$ (validation set only, where $L^{[2]}$ degenerates to $L^{[1]}$). This is expected, because $L^{[2]}$ "wasted" some sample on its cross-validation formulation. Intuitively, the variance (or CI) can be described by the curvature of $L$ at maximum. Note that from $L^{[2]} = L(\Theta; D_\tau^{tr}, D_\tau^{eval}) - L(\Theta; D_\tau^{tr})$, part of the curvature is cancelled out by the existence of second term, resulting in a larger variance of $\widehat{\Theta}$.

### A.3. Bound of GEM gradient estimation error (*Section 3.2*)

We show a general proposition in VI (or other measure approximation methods). Define

$$g(x) = \mathrm{E}_{Z \sim P}\, f(x, Z) \tag{10}$$
$$\tilde{g}(x) = \mathrm{E}_{Z \sim Q}\, f(x, Z) \tag{11}$$

To make $\tilde{g}(x) \approx g(x)$, we let $Q \approx P$, in the sense that $D_{\mathrm{KL}}(Q\|P)$ is minimized over $Q$. Then

$$\|g(x) - \tilde{g}(x)\| = \left\|\mathrm{E}_{Z \sim P}\left[f(x, Z)\left(1 - \frac{q(Z)}{p(Z)}\right)\right]\right\| \tag{12}$$

$$\leq \mathrm{E}_{Z \sim P}\left[\|f(x, Z)\| \cdot \left|1 - \frac{q(Z)}{p(Z)}\right|\right] \tag{13}$$

$$\leq \left[\mathrm{E}_{Z \sim P}\|f(x, Z)\|^2\right]^{\frac{1}{2}} \cdot \left[\mathrm{E}_{Z \sim P}\left|1 - \frac{q(Z)}{p(Z)}\right|^2\right]^{\frac{1}{2}} \tag{14}$$

$$\leq M \cdot \mathrm{E}_{Z \sim P}\left|1 - \frac{q(Z)}{p(Z)}\right| \tag{15}$$

$$= M \cdot \int |p(z) - q(z)|\, dz \tag{16}$$

$$= 2M \cdot D_{\mathrm{TV}}(P, Q) \quad \text{(definition of totoal variation)} \tag{17}$$

$$\leq \sqrt{2}M \cdot \sqrt{D_{\mathrm{KL}}(Q\|P)} \quad \text{(Pinsker's inequality)} \tag{18}$$

where $M = \left[\mathrm{E}_{Z \sim P}\|f(x, Z)\|^2\right]^{\frac{1}{2}}$. In our discussion of gradient approximation, we let $x = \Theta$, $z = \theta$, $f(x, z) = \nabla_\Theta \log p(\theta|\Theta)$, and $p(z) = p(\theta|D, \Theta)$. Then $M = \left[\mathrm{E}_{\theta \sim p(\theta|D, \Theta)}\|\nabla_\Theta \log p(\theta|\Theta)\|^2\right]^{\frac{1}{2}}$

### A.4. $L^{[2]}$ property (*Section 3.2*)

**Property 1.** $L(\Theta, D_\tau^{tr}; D_\tau^{val}) = L(\Theta; D_\tau^{tr} \bigcup D_\tau^{val}) - L(\Theta; D_\tau^{tr})$

*Proof.*

$$L(\Theta, D_\tau^{tr}; D_\tau^{val}) = \log P(D_\tau^{val}|\Theta, D_\tau^{tr})$$

$$= \log \int P(D_\tau^{val}|\theta_\tau) * P(\theta_\tau|\Theta, D_\tau^{tr}) d\theta_\tau$$

$$= \log \int P(D_\tau^{val}|\theta_\tau) * \frac{P(D_\tau^{tr}|\theta_\tau)P(\theta_\tau|\Theta)}{P(D_\tau^{tr}|\Theta)} d\theta_\tau$$

$$= \log \int \frac{P(D_\tau^{val\oplus tr}|\theta_\tau)P(\theta_\tau|\Theta)}{P(D_\tau^{tr}|\Theta)} d\theta_\tau$$

$$= \log \frac{P(D_\tau^{val\oplus tr}|\Theta)}{P(D_\tau^{tr}|\Theta)}$$

$$= L(\Theta; D_\tau^{tr} \bigcup D_\tau^{val}) - L(\Theta; D_\tau^{tr})$$

$\square$

## B. Theory

### B.1. generative model of RL (*Section 2.2*)

Using the relation between posterior and ELBO we have $P(\theta|D; \Theta) = \arg\max_g E_{\theta \sim g} \log P(D|\theta) - D_{KL}(g|\Theta) = \arg\min E_{\theta \sim g}\mathcal{L}(D, f_\theta) + D_{KL}(g|\Theta)$. In RL, D is trajectories $\{x_t, a_t, r_t\}_{t=1}^H$. In policy gradient, $\pi(a|x) = f_\theta(x)(a)$, so $\mathcal{L}(D, f_\theta) = \sum_t \pi(a_t|x_t) * r_t = \sum_t f_\theta(x_t)(a_t) * r_t$. The posterior tends to find distribution of $\theta$ that maximize the expected loss function under regularization of a KL-distance to the prior. For a given environment, when data is infinitely sufficient (at least sufficient $\{x_t, a_t, r_t\}$ tuples those appear in the optimal policy MDP), the posterior goes to a delta distribution of the optimal policy.

### B.2. non-uniqueness, fast-adaptation, $A^\star$ and Gaussian (*Section 2.3*)

For neural networks $f(; \theta)$, there exist many local minimums that have the similar good performance. For each task, our objective is to find any one of them instead of the only true optimal among them. Inspired by this important observation, we model the case as non-uniqueness where more than one best parameter $\theta_\tau$ exist for each task $\tau$. We denote the set of best parameters for task $\tau$ as $\{\theta_{\tau_i}(\tau)\}_{\tau_i=1}^{n_\tau}$, where $n_\tau$ is the number of coexisting best parameters for task $\tau$. If we choose any one of the best parameters for each task and form a set $\{\tau_i\}$ we can get a corresponding distribution $\theta_{\tau_i}(\tau) \sim P_{\{\tau_i\}}(\theta_\tau)$ induced by $P(\tau)$ (change of variable). There are $\prod n_\tau$ choices of sets and the same number of distributions $P_{\{\tau_i\}}(\theta_\tau)$ denoted as $\mathcal{P}$. Theorem 1 holds for any distribution in $\mathcal{P}$ which means we can use any of them as prior to come up with optimal decision rules at meta-testing.

However, it's not easy to model an arbitrary prior distribution with effective and efficient Bayesian inference. The common feasible Bayesian Inference method for neural networks is gradient based variational inference with Gaussian parametric approximation. This method only works well for distributions that are uni-modal with small variance or multi-modal and each with small variance(which can be modeled by mixture Gaussian) for two reasons. First, the smaller the variance of the distribution the lower the approximate error of Gaussian(property of Gaussian approximation). Second, prior $P$ also serves as initial points in the fast-adaptation Bayesian inference procedure. This requires $P_{\{\tau_i\}}(\theta_\tau)$ to be compact enough such that each task posterior can be attained within a few variational inference gradient steps.

For single cluster of tasks, we show empirical evidences in Appendix C that there exist such kind of a distribution. In another word, there exist a small neighbouring area $A^\star$ where most tasks have at least one best parameter inside it as shown in Figure 1(a). Some other works about multi-modal meta-learning also provide evidences for the feasibility of applying mixture Gaussian to multi-cluster tasks situation which our methods can be adapted to (Grant et al., 2018b; Rasmussen, 2000). In this work we focus on the uni-modal situation and leave the multi-modal situation to future work. So in this work, we use Gaussian $P(\theta; \Theta)$ in the above framework. By doing so $P(\theta; \Theta)$ will converge to the best fit of the smallest variance distribution in $\mathcal{P}$ because Gaussian fits the smallest variance distribution best.

## B.3. co-ordinate descent (*Section 3.2*)

Following the ELBO property mentioned in Section 3.2 we have

$$\max_{\Theta} L(\Theta; D_\tau) \simeq \max_{\Theta} E_{P(\theta_\tau; \lambda_\tau^*(\Theta))}[\log P(D_\tau, \theta_\tau | \Theta) - \log P(\theta_\tau; \lambda_\tau^*(\Theta))]$$

$$= \max_{\Theta} E_{P(\theta_\tau; \lambda_\tau^*(\Theta))}[\log P(D_\tau | \theta_\tau) + \log P(\theta_\tau | \Theta) - \log P(\theta_\tau; \lambda_\tau^*(\Theta))]$$

$$= \max_{\Theta}[E_{P(\theta_\tau; \lambda_\tau^*(\Theta))} \log P(D_\tau | \theta_\tau)] - KL(P(\theta_\tau; \lambda_\tau^*(\Theta)) \parallel P(\theta_\tau | \Theta)) \qquad (19)$$

$$= \max_{\Theta} ELBO^{(\tau)}(\lambda_\tau^*(\Theta), \Theta)$$

$$= \max_{\Theta, \lambda_\tau}[E_{P(\theta_\tau; \lambda_\tau)} \log P(D_\tau | \theta_\tau)] - KL(P(\theta_\tau; \lambda_\tau) \parallel P(\theta_\tau | \Theta)) \qquad (20)$$

Now we can show that algorithm GEM-BML is an stochastic co-ordinate descent algorithm to optimize ELBO and thus optimize $L^{[1]} = E_\tau L(\Theta; D_\tau)$. For each iteration we sample a batch of tasks $\tau$ and optimize over $\lambda_\tau$ and $\Theta$ alternately. At inner-update, we fix $\Theta$ and maximize (20) in terms of $\lambda_\tau$, $\lambda_\tau \leftarrow \arg\max_{\lambda_\tau} ELBO^{(\tau)}(\lambda_\tau, \Theta)$ which corresponds to the posterior computation in Line 2,3 of Subroutine `GEM-BML`. At meta-update, we fix $\lambda_\tau$ and improve (20) in terms of $\Theta$, $\Theta \leftarrow \Theta - \beta \nabla_\Theta KL(P(\theta_\tau; \lambda_\tau) \parallel P(\theta_\tau | \Theta)) = \Theta - \beta E_{\theta \sim P(\theta; \lambda_\tau)} \nabla_\Theta \log[P(\theta; \Theta)]$ which corresponds to the $\Theta$ update in Line 4 of Subroutine `GEM-BML` and Line 10 of Algorithm 1.

## B.4. recasting related works to our framework (*Section 4*)

For simpicity, we first set up some notations as follows:
sg: stop gradient
$D_2 = D_\tau^{eval}$
$D_1 = D_\tau^{tr}$
$\lambda_2(\Theta)$: trained posterior given $D_2 \bigcup D_1$
$\lambda_1(\Theta)$: trained posterior given $D_1$
$L_i(\Theta)$: $E_{P(\theta_\tau; \Theta)} \log P(D_i | \theta_\tau)$
$L_{i,j}^{pos}(\Theta)$: $E_{P(\theta_\tau; \lambda_i(\Theta))} \log P(D_j | \theta_\tau)$, the gradients of which are estimated by LRP or Flipout (Kingma et al., 2015; Zhang et al., 2018).
$L_i^{pos} = L_{i,i}^{pos}$

Recall that $L(\Theta; D_\tau) \simeq ELBO^{(\tau)}(\lambda_\tau(D_\tau; \Theta); \Theta) = [E_{P(\theta_\tau; \lambda_\tau(D_\tau; \Theta))} \log P(D_\tau | \theta_\tau)] - KL[P(\theta_\tau; \lambda_\tau(D_\tau; \Theta)) \parallel P(\theta_\tau | \Theta)]$. Apply ELBO gradient estimator to $L_\tau^{[1]} = L(\Theta; D_1)$ we get $\nabla_\Theta[L_1^{pos}(\Theta) - KL(\lambda_1(\Theta), \Theta)]$ which is the meta-gradient of Amortized BML. In the original work they have a variant of $\nabla_\Theta[L_{1,2}^{pos}(\Theta) - KL(\lambda_1(\Theta), \Theta)]$ to improve the generalization. Apply ELBO gradient estimator to $L_\tau^{[2]} = L(\lambda_\tau(D_1; \Theta); D_2)$ (simply replace $\Theta$ of $\lambda_\tau(D_1; \Theta)$) we get $\nabla_\Theta[L_2^{pos}(\Theta) - KL(\lambda_2(\Theta), \lambda_1(\Theta))]$ which corresponds to the meta-gradient of PMAML.

Apply GEM gradient estimator $\hat{g} = E_{\theta_\tau \sim P(\theta_\tau; \lambda_\tau(D_\tau, \Theta))} \nabla_\Theta \log[P(\theta_\tau; \Theta)]$ to $L_\tau^{[1]}$ and $L_\tau^{[2]}$ as the above procedure we get GEM-BML: $\nabla_\Theta[KL(sg(\lambda_1(\Theta)), \Theta), KL(sg(\lambda_2(\Theta)), \Theta)]$ and KL-Chaser Loss: $\nabla_\Theta[KL(sg(\lambda_2(\Theta)), \lambda_1(\Theta))]$. The Chaser Loss meta-gradient in BMAML is $\nabla_\Theta[\parallel sg(\lambda_2(\Theta)) - \lambda_1(\Theta) \parallel^2]$ which is similar to KL-Chaser Loss but replace the KL loss with $l_2$ loss.

## B.5. Advantages of our methods (*Section 4*)

Observe that all methods in the above matrix requires to compute the posterior parameters $\lambda_\tau(D_\tau; \Theta)$ first and use it to compute the sampled meta-loss function gradient $\nabla_\Theta L_\tau^{[i]}$. Following the convention of (Finn et al., 2017), we define the step of computing $\lambda_\tau(D_\tau; \Theta)$ as *inner-update* and the step of computing $\nabla_\Theta L_\tau^{[i]}$ as *meta-update*. Notice that both the $L^{[2]} = E_{\tau \in} L(\lambda_\tau(D_\tau^{tr}; \Theta); D_\tau^{val})$ column and the ELBO-gradient involves the computation of $\nabla_\Theta \lambda_\tau(D_\tau; \Theta)$. This means the inner-update computation of these three methods(highlighted in colour) has to be built in Tensors in order to compute the gradients by auto-grad. This tensor building and backProps procedure has several drawbacks. First, this procedure is time-consuming, if using SGD for inner-update, the computation time grows rapidly as the number of inner-update gradient steps increase as we show in Experiment (Figure 2), which limits the number of maximum steps. Empirical evidence is

provided in C.2 where multiple inner-update gradient steps are necessary for this framework of methods to work. Second, this procedure limits the choice of optimization method in inner-update. The only optimization method so far that can be trivially written in Tensors is SGD. However, in many situations SGD is not enough or sub-optimal for this framework to work. We show policy-gradient RL examples in Experiment where Trust Region optimization instead of SGD in inner-update is necessary and supervise learning examples where ADAM optimizer works better than SGD. Our method, on the other way, avoids the computation of $\nabla_\Theta \lambda_\tau(D_\tau; \Theta)$ and thus avoids the drawbacks mentioned above. Since it only requires the value of inner-update result $\lambda_\tau(D_\tau; \Theta)$ instead of the Tensor of optimization process, the meta-update and inner-update can be decoupled. For inner-update, it has much more degree of freedom in choosing optimization methods to compute the value of $\lambda_\tau(D_\tau; \Theta)$ without the burden of building Tensors on optimization process as mentioned above. For meta-update, it avoids back-propagation computations and does not involve with data explicitly since it only requires the value of $\lambda_\tau(D_\tau; \Theta)$ to compute meta-gradient (Line 4 of Subroutine `GEM-BML` and `GEM-BML+` ). This gives our method more potential for distributed computing and privacy sensitive situations. Also notice that, if assuming independence between neural network layers, the GEM-gradient can be computed among different neural network layers in parallel, which may largely reduce the computation time in deep neural networks.

### B.6. Other methods to compute Meta-Gradient (*Section 3.2*)

There are several other methods of Subroutine `Meta-Gradient` in previous works. (Grant et al., 2018a) uses Gaussian $P(\theta|\Theta)$ and approximate $P(D_\tau|\theta)$ with Gaussian by applying Laplace approximation which uses a second-order Taylor expansion of $P(D_\tau|\theta)$. However, there are evidences show that for neural network $P(D_\tau|\theta)$ can be highly asymmetric. Approximate it with symmetric distribution such as Gaussian may cause a series of problem. (Yoon et al., 2018) proposes to use M particles $\Theta = \{\theta^m\}_{m=1}^M$ to represent $\theta \sim P(\theta|\Theta)$ and compute gradients on them $\nabla_\Theta L(\Theta; D_\tau) = \nabla_{\{\theta^m\}_{m=1}^M} \log[\frac{1}{M} \sum_{m=1}^M P(D_\tau|\theta^m)]$. This methods requires $O(M^2)$ times more computation in each gradient iteration.

### B.7. Gaussian case solution (*Section 3.2*)

Under Gaussian approximation, we assume the prior and approximate posteriors to be $P(\theta_\tau|\Theta) \sim N(\mu_\Theta, \Lambda_\Theta^{-1})$ and $q(\theta_\tau; \lambda_\tau^{tr}) \sim N(\mu_{\theta_\tau}^{tr}, \Lambda_{\theta_\tau}^{tr})$, $q(\theta_\tau; \lambda_\tau^{tr\oplus val}) \sim N(\mu_{\theta_\tau}^{tr\oplus val}, \Lambda_{\theta_\tau}^{tr\oplus val})$. Then the meta-gradient of GEM-BML+ $\nabla_\Theta L(\Theta, D^{tr}; D^{val})$ has close form solution given as follows.

$$
\begin{aligned}
\frac{\partial L(\Theta, D^{tr}; D^{val})}{\partial \mu_\Theta} &= \sum_{\tau \in} (\mu_{\theta_\tau}^{tr\oplus val} - \mu_{\theta_\tau}^{tr})^T \Lambda_\Theta^{-1} \\
\frac{\partial L(\Theta, D^{tr}; D^{val})}{\partial \Lambda_\Theta^{-1}} &= \sum_{\tau \in} -\frac{1}{2}(\Lambda_{\theta_\tau}^{tr\oplus val} - \Lambda_{\theta_\tau}^{tr}) \\
&\quad - \frac{1}{2}(\mu_{\theta_\tau}^{tr\oplus val} - \mu_{\theta_\tau}^{tr})(\mu_{\theta_\tau}^{tr\oplus val} + \mu_{\theta_\tau}^{tr} - 2\mu_\Theta)^T
\end{aligned}
\tag{21}
$$

## C. Experiment

### C.1. Meta-Gradient estimation error (*Section 3.2*)

To study the question of meta-gradient accuracy, we considers a synthetic lineare regression example. This provides an analytical expression for the true meta-gradient $\nabla_\Theta L(\Theta; D_\tau)$, allowing us to compute the estimation error of different `Meta-Gradient` subroutines. We plot in Figure 1 the estimation error of repeated random runs. We find that both GEM and ELBO-gradient asymptotically match the exact meta-gradient, but GEM computes a better approximation in finite iterations with more stability.

### C.2. Necessity of many inner-update steps example (*Section 1*)

This example is based on the same sinusoidal function regression problem in Section 5.1 with a slightly easier setting than the challenging one. All the settings are the same as described in Section 5.1 except the noise parameter $A = 0$ and $\omega \in [0.5, 1.0]$. We plot in Figure 2 the meta-test result for MAML with number of inner-update equals to 1,2,3. We can see clearly that in this case multiply inner-update steps is necessary and important for MAML to work. We observe similar

*Figure 1.* Meta-Gradient estimation error

*Figure 2.* Necessity of many inner-update steps example

phenomena for other methods under our extended EB framework.

### C.3. A* (*Section 2.3*)

We use MAML for this experiment on sinusoidal function regression with the default setting and image classification on Omniglot. Let $\Theta$ be a well learned initial point(delta prior) by the meta-train process. At meta-test, denote $\theta_i$ as the adapted parameter(delta posterior) from $\Theta$ on a new task $i$. We have verified that MAML works on the two settings we use for this experiment in the sense that $L(D_i; \theta_i) \ll L(D_i; \Theta)$. Now we provide evidence that the area enclosed by $\{\theta_i\}, i \sim P(\tau)$ is a small neighboring area $A*$ that any point within it is a good initial point with good meta-test behaviour. To be specific, we randomly choose any point within the convex combination of $\{\theta_i\}, i \in \mathcal{T}_{meta-test}$ as initial point $\Theta$, then use it as initial point for meta-test on new tasks. We plot in Figure 3 the meta-test result of some random runs and the average of 100 random runs. We can see that the original trained initial point does have the best performance but other random initial points within $A*$ also have good performance. We also observe that random initial points within $A^*$ has lower error before adaptation while losing some fast adaptation ability. This is consistent with our intuition that MAML tends to find a point in the center of $A^*$ which has the best few-step reach-out ability to all task parameters within $A^*$. While a random initial points within $A^*$ may be close to some of the task parameters and a little bit more far away from other task parameters.

### C.4. Experiment Details

We summarize the hyperparameters in Table 1, 2 and 3, in which Meta-batch size is the number of tasks used in one meta-update iteration. All experiments were conducted on a single NVIDIA (Tesla P40) GPU.

*Figure 3.* $A^\star$

### C.4.1. REGRESSION

All comparing models are trained using the same network architecture and initialized with the same parameters. For all models, the negative log likelihood $-\log P(D_\tau | \theta_\tau)$ is the mean squared error between the predicted and true y value and the same for loss functions in all other models.

| | |
|---|---|
| Meta-update Learning Rate | 0.001 |
| Inner-update Learning Rate | 0.001 |
| Inner Gradient steps at meta-train | 1 |
| Inner Gradient steps at meta-test | 10 |
| Meta-batch Size | 5 |

*Table 1.* Hyperparameters for sinusoidal regressions.

### C.4.2. CLASSIFICATION

The set up of N-way classification is as follows: select N unseen classes, provide the model with 1 or 5 different instances of each of the N classes, and evaluate the model's ability to classify new instances within the N classes. For Omniglot, 1200 characters are selected for training, and the remaining are used for testing, irrespective of the alphabet. Each of the characters is augmented with rotations by multiples of 90 degrees (Santoro et al., 2016). Our Bayesian Neural Network follows the same architecture as the embedding function used by (Finn et al., 2017), which has 4 modules with $3 \times 3$ convolutions and 64 filters, followed by batch normalization ((Ioffe & Szegedy, 2015)), a ReLU non-linearity, and $2 \times 2$ max-pooling. The Omniglot images are downsampled to $28 \times 28$, so the dimensionality of the last hidden layer is 64. The last layer is fed into a softmax (Vinyals et al., 2016). For Omniglot, we used strided convolutions instead of max-pooling. For MiniImagenet, we used 32 filters per layer to reduce overfitting. For all models, the negative log likelihood $-\log P(D_\tau | \theta_\tau)$ is the cross-entropy error between the predicted and true class.

| | Omniglot,5-class | Omniglot,5-class | miniImageNet |
|---|---|---|---|
| Meta-update Learning Rate | 0.001 | 0.001 | 0.001 |
| Inner-update Learning Rate | 0.01 | 0.01 | 0.001 |
| Inner Gradient steps at meta-train | 1 | 5 | 5 |
| Inner Gradient steps at meta-test | 10 | 10 | 10 |
| Meta-batch Size | 32 | 16 | 4 |

*Table 2.* Hyperparameters for few-shot image classifications.

## C.4.3. REINFORCEMENT LEARNING

In 2D Navigation, the point agent is trained to move to different goal positions in 2D, randomly chosen for each task within a unit square. The observation is it current position, and actions are velocity clipped to be in the range $[-0.1, 0.1]$. The reward is the negative squared distance to the goal. For MuJoCo continuous control, we perform goal velocity and goal direction two kinds of task on half cheetah (our available current infrastructure is limited to perform experiment on more advanced environment like 3D ant, we leave it to future work). In the goal velocity task, the agent receives higher rewards as its current velocity approaches the goal velocity of the task. In the goal direction task, the reward is the magnitude of the velocity in either the forward or backward direction. The goal velocity is sampled uniformly at random from $[0.0, 2.0]$ for the cheetah.

|  | 2D navigation | half-cheeah, goal velocity | half-cheeah, forward/backward |
|---|---|---|---|
| Meta-update Learning Rate | 0.001 | 0.001 | 0.001 |
| Inner-update Learning Rate | 0.1 | 0.1 | 0.1 |
| Inner Gradient steps at meta-train | 1 | 1 | 1 |
| Inner Gradient steps at meta-test | 3 | 3 | 3 |
| Meta-batch Size | 20 | 40 | 40 |
| Inner-batch Size | 20 | 20 | 20 |

*Table 3.* Hyperparameters for reinforcement learning.