[Reviews · NeurIPS 2020]

Review 1

Summary and Contributions: This paper proposed a new Bayesian Meta-learning approach inspired by Gradient-EM. The meta-update step only relies on the solutions of the inner-update step instead of the optimization process and data, so it achieves high efficiency compared with other second-order gradient-based meta-learning approaches.

Strengths: 1. New bayesian meta-learning approach with high efficiency in computation. 2. Excellent theoretical intepretation. 3. Impressive performance in existing gradient-based meta-learning methods.

Weaknesses: The work is not very insightful to the community. Improving the computation efficiency of grandient-based meta-update has been considerably stuided in recent years. To this, the proposed method just lacks the necessary comparison with other baselines (like implicit MAML) in efficiency. The authors also claim their method can be easily turned into parallel version (see ''Unlike MAML ...... in Algorithm 1''), however, did not verify this claim in their experiment. Does the parallelization lead to low performance?

Correctness: Yes.

Clarity: Well written. The author propose new method and further explain its theoretical property.

Relation to Prior Work: The paper uses a short related work section to discuss the difference between GEM-BML and other previous meta-learning algorithm. I think it could be extended

Reproducibility: Yes

Additional Feedback: It seems the algorithm is quite easy to implement and reproduce the authors' empirical results, while it would be more reassuring if they can provide the code and model.


Review 2

Summary and Contributions: The authors propose an extension to classic model agnostic meta-learning (MAML) via a gradient-EM approach, whose key contribution is the decoupling of the inner gradient steps and the meta-update steps during meta training. Most of the meta learning approaches based on hierarchical Bayes can be recast in this formulation

Strengths: The proposed methodology casts classic MAML in a Bayesian framework (which is not entirely new as a concept) and additionally provides a technique for decoupling meta-update and inner update, together with a thorough analysis of why it mathematically holds.

Weaknesses: While the method provides an improvement to existing Bayesian MAML, the contribution is somewhat minor. The theoretical foundations of Bayesian meta-learning have been studied in most of the cited literature, which then leads to some questions regarding the claims that are considered as novel, while potentially already discussed in the previous literature. Furthermore I find confusing the idea in Figure 1a, discussed in section 2.3: what is this set and what are the authors claiming exactly?

Correctness: The method seems correct to the best of my knowledge.

Clarity: Unfortunately I had a very hard time understanding the paper. I would strongly suggest major English revision. Also, please add the reference to the Appendix with the corresponding letter.

Relation to Prior Work: The related literature is discussed, however the differences are often unclear to understand: I believe some of the claims in this paper were already (at least in part) discussed in previous literature.

Reproducibility: Yes

Additional Feedback: Please check the format of your supplementary material as it refers to ICML 2019. -------------------------------------- EDIT ------------------------------------------------------------- The author response addressed some essential concerns raised in the reviewing process. While the ideas are there I still think that the paper should undergo major reviewing and polishing to strengthen the results and the novelty, motivating a score raise to 5.


Review 3

Summary and Contributions: This paper proposes a new Bayesian meta-learning method that is computationally efficient by decoupling inner-update and meta-update. And it shows better performance than baselines (MAML, Reptile) with less computation usage.

Strengths: As I know, it is first to decouple inner-update and meta-update on Bayesian MAML families. It clearly shows the benefit of efficient learning in the experiment section.

Weaknesses: It proposes more efficient Bayesian meta-learning method but doesn't compare and analyze the performance with other more complex Bayesian meta-learning methods.

Correctness: Correct

Clarity: Too dense to read some sections (e.g., section 3.2 or 4)

Relation to Prior Work: Yes

Reproducibility: Yes

Additional Feedback: - Many equations are in paragraphes, which cause hard to read. I think that making those equations as separate lines with description is helpful to make this paper more easy to read. - Comparison with PMAML or BMAML can be interesting even only for simple tasks like regression. - Fig 2 (b) resolution is too low to check the legend and x-axis legend is not well captured. Bellow comments are about typos or other minor things. - You can refer appendix on main paper, by splitting pdf file after making the pdf from tex. - line 149 in Theorem 2, \tau suddenly appeares and removes during. Is it right? - line 186, one the other hand -> on the other hand - line 298, SVDG -> SVGD. - line 236, in Fig2 (b), you didn't compare with BMAML. # To authors: Thank you for updating the results with BMAML. It makes your proposal more clearly (less computation, better performance).


Review 4

Summary and Contributions: This paper proposes a gradient-EM based Bayesian meta-learning framework. The key ingredient is using the gradient approximation technique similar to the M-step of EM algorithm for the efficient computation of meta-gradient (the gradient of the meta-level parameter \Theta). The gradient EM approach does not require the backpropagation through the inner update step which typically requires the computation of Hessian or even higher-order derivatives so limits the scalability of meta-learning algorithms. The proposed algorithm is demonstrated to be efficient (takes less GPU memory and computation time) yet accurate than the baselines on synthetic regression, image classification, and reinforcement learning tasks.

Strengths: - Provides a justification not having to backpropagate through the inner-updates (i.e., through \lambda(D_\tau;\Theta)) in the context of EM-like procedure. - A nice summary by comparing and recasting the proposed framework to the existing meta-learning algorithms (Table 2, Section B.4 in the appendix). - Some theoretical guarantees (Section A.2., A.3. in the appendix).

Weaknesses: - Technical novelty is somewhat limited. - Weak experimental results. - The analysis (asymptotic normality, bound on the GEM error) is not really helpful (too general). - Presentation should be enhanced.

Correctness: I think the general argument presented in the paper is correct. Most of the theorems are already known results or derived easily from already known results.

Clarity: No, the paper uses many misleading notations without or not very explicitly introducing them. There seem to be quite a lot of typo. To list a few, - Line 62-64 is quite confusing. What does 2K samples correspond to? D_\tau^(meta-train) + D_\tau^(meta-test)? Is tau for meta-train and meta-test is different? - in line 89, L^[1]_tau should be L(\Theta; D_\tau) instead of L(\Theta; D_\tau^eval) - In line 152, did you mean Theorem 2 instead of Theorem 1? - Many symbols are used for several different notions. \tau is typically used for the tasks, but in line 69 it is used as \tau(\theta) to denote a generative model. - In the legend of the first plot of Figure 2, ELBO should be amortized BML?

Relation to Prior Work: The paper is doing a good job of comparing the proposed method to existing works, by first introducing their general framework and recasting some of them as special cases or variants (using different types of losses). However, those discussions are limited to relatively old methods, and does not include more recent gradient-based meta-learning baselines.

Reproducibility: Yes

Additional Feedback: I think the technical novelty of the proposed algorithm is somewhat limited since the resulting algorithm is essentially a Bayesian version of reptile (GEM-BML using L^[1] loss). Nevertheless, I like the reinterpretation given in the paper, especially the co-ordinate decent view of meta-update decoupling the inner-level update and outer-level update. Any argument highlighting the technical novelty of the proposed method would be appreciated. The important aspect of the proposed method is its robustness due to being Bayesian. I think the paper could be strengthened by including more experiments to see this aspect by testing performance or calibration under distributional shift. The performance of GEM-BML and GEM-BML+ is not that impressive for typical few-shot learning settings (the difference with baselines are not significant in a statistical sense for some settings).

[Author Response · NeurIPS 2020]



**Contribution I: General Bayesian Framework** Our work is motivated by a general question: what is the optimal
solution of meta learning, or how can we extract information from known tasks to help the most in the new tasks? Our
approach started with a generalized meta learning setting in 2.1, where data in meta-training phase is not artificially
divided into training/testing as in previous work. Then we provided Theorem 1 as the foundation of our general
Bayesian framework by claiming its optimality under certain metrics. We agree with Reviewer #4, that the mathematical
techniques are easily derived, nonetheless, the theory on $L^{[2]}$ method has not been established in the field of meta
learning to the best of our knowledge (e.g. empirical Bayes focuses on $L^{[1]}$ with fixed number of tasks). As shown, $L^{[2]}$
is essential for NN based models, especially in RL, which highlighted our contribution of GEM-BML+ algorithm.

**Contribution II: EM Style Algorithm** Our general Bayesian framework inspires EM style (or coordinate descent,
as Reviewer # 4 pointed out) algorithms where meta-update only depends on the solution of each inner-update. This
is in contrast to MAML style where meta-update depends on the process of each inner-update (path dependence).
This is an important contribution because it brings huge potential on the improvement of efficiency, scalability and
flexibility compare to MAML style algorithm (Line 37-50, Appendix B5). Side note on iMAML: this gradient-based
meta-learning method is not within our framework though it shares the EM style. We didn't think it's important to
include iMAML in performance comparison because its EM style update is by the effort of approximating MAML rather
than a Bayesian approach. Also, it requires Hessian as second-order information and solving QP at every meta-update
step (Line 303).

**Contribution III: Summary and Future Work** As recognized, we provided a nice summary on a variety of related
work as in Table 1. However, one major contribution is that Table 1 can be extended with more columns (e.g. one more
column of $L^{[2]} = E_{\tau \in \mathcal{T}} L(\Theta; D_\tau^{tr} \cup D_\tau^{val}) - L(\Theta; D_\tau^{tr})$) and more rows to a larger matrix with some blank elements
(models) that haven't been explored before. For example, KL-Chaser Loss model in the right bottom of Table 1 hasn't
been studied before. We leave the thorough study of all combinations to future work. This kind of recasting/summary
is novel to the best of our knowledge. [E.Grant 2018 Recasting] provided a qualitative recasting of MAML to EB
using the loose connection between early stopping and implicit prior, which is totally a different story from our work.
Amortized BML directly goes into ELBO gradient and BMAML proposes "learn to infer". We hope this work not only
provides clear overview of methods in meta learning, but also sheds light on future work.

*Responses to specific questions*
Reviewer # 1: a) We did compare with iMAML in Figure 2(see also the above Figure) b) If we assume independent
distribution of neural weights (Line 203) between NN layers (which is the case in our experiments and almost all
implementations of Bayesian NN), Line 4 of GEM-BML(+) can be written as the sum of different layers and therefore
the parallelization will give exactly the same results as our experiment with a reduction in computational time. c) A
large portion of "related work" is actually within the body(Section 2,3,4 and Appendix B6).
Reviewer # 2: For Figure 1(a), each color refers to a task and each point is regarding a set of model weights (NN
weights) which has good performance on that task. This figure demonstrates that many good solutions exist for each
task (local optimums of NN,Line 99). The dotted line area is the small neighboring zone $A^*$ where each color has at
least one point inside (Section 2.3 and Appendix B.2 provide further explanation).
Reviewer # 3: We choose Amorized BML as the benchmark of Bayesian meta-learning to com-
pare(typo:ELBO=>Amortized BML) because its performance is comparable to all other previous Bayesian meta-
learning algorithms. We actually included BMAML in a previous version of our paper(See the Figure above for both
performance(left two,blue) and efficiency(right two,red)). The efficiency of BMAML is not within feasible range. As
for PMAML, the code is not released and the algorithm is complex to reproduce.
Reviewer # 4: We realized that "GEM is the Bayesian version of Reptile" can mislead to a trivial impression on our
work. We would like to point out that, our method was not motivated by "how can we improve from Reptile" (like from
MAML to BMAML), but from a total different perspective. The connection with Reptile was a coincidence discovery
(while we were working on the recast/summary). Lastly, Reptile can not be applied to RL (as stated by its author and
confirmed in our experiments) while our method performs excellently.

[Meta-Review · NeurIPS 2020]

This paper makes a solid contribution to the meta-learning area that is computational efficient. The reviewers also praised the execution and were largely convinced by the reported results. The authors also clarified the concerns raised by the reviewers in their rebuttal. The presentation could be improved, but is acceptable.